# Prevalence, Characteristics, Management and Outcomes of Patients with Heart Failure with Preserved, Mildly Reduced, and Reduced Ejection Fraction in Spain

**DOI:** 10.3390/jcm11175199

**Published:** 2022-09-02

**Authors:** Carlos Escobar, Beatriz Palacios, Luis Varela, Martín Gutiérrez, Mai Duong, Hungta Chen, Nahila Justo, Javier Cid-Ruzafa, Ignacio Hernández, Phillip R. Hunt, Juan F. Delgado

**Affiliations:** 1Cardiology Department, University Hospital La Paz, 28046 Madrid, Spain; 2AstraZeneca Farmaceutica, 28033 Madrid, Spain; 3Evidera, London W6 8BJ, UK; 4AstraZeneca, Gaithersburg, MD 20878, USA; 5Evidera, 113 21 Stockholm, Sweden; 6Karolinska Institute, Department of Neurobiology, Care Sciences, and Society, 171 77 Stockholm, Sweden; 7Evidera, 08005 Barcelona, Spain; 8Atrys Health, 28001 Madrid, Spain; 9Cardiology Department, University Hospital 12 de Octubre, CIBERCV, 28041 Madrid, Spain

**Keywords:** cardiovascular, heart failure, sacubitril/valsartan, SGLT2 inhibitors

## Abstract

Objective: To estimate the prevalence, incidence, and describe the characteristics and management of patients with heart failure with preserved (HFpEF), mildly reduced (HFmrEF), and reduced ejection fraction (HFrEF) in Spain. Methods: Adults with ≥1 inpatient or outpatient HF diagnosis between 1 January 2013 and 30 September 2019 were identified through the BIG-PAC database. Annual incidence and prevalence by EF phenotype were estimated. Characteristics by EF phenotype were described in the 2016 and 2019 HF prevalent cohorts and outcomes in the 2016 HF prevalent cohort. Results: Overall, HF incidence and prevalence were 0.32/100 person-years and 2.34%, respectively, but increased every year. In 2019, 49.3% had HFrEF, 38.1% had HFpEF, and 4.3% had HFmrEF (in 8.3%, EF was not available). Compared with HFrEF, patients with HFpEF were largely female, older, and had more atrial fibrillation but less atherosclerotic cardiovascular disease. Among patients with HFrEF, 76.3% were taking renin-angiotensin system inhibitors, 69.5% beta-blockers, 36.8% aldosterone antagonists, 12.5% sacubitril/valsartan and 6.7% SGLT2 inhibitors. Patients with HFpEF and HFmrEF took fewer HF drugs compared to HFrEF. Overall, the event rates of HF hospitalization were 231.6/1000 person-years, which is more common in HFrEF patients. No clinically relevant differences were found in patients with HFpEF, regardless EF (50- < 60% vs. ≥60%). Conclusions: >2% of patients have HF, of which around 50% have HFrEF and 40% have HFpEF. The prevalence of HF is increasing over time. Clinical characteristics by EF phenotype are consistent with previous studies. The risk of outcomes, particularly HF hospitalization, remains high, likely related to insufficient HF treatment.

## 1. Introduction

More than 60 million people worldwide have heart failure (HF) [1,2]. However, these numbers will increase in the following years, mainly due to the elderly of the population, but also because of the higher prevalence of some comorbidities, such as hypertension or diabetes, and the better treatment of acute cardiovascular conditions [1,2,3,4,5]. Thus, it has been projected in some European countries that the prevalence of HF will increase by 30% in 2035 [6]. In Spain, in 2019, there were more than 750,000 adults with HF, with an annual incidence of 2.78/1000 individuals/year [7].

The use of recommended HF drugs in clinical practice reduces the HF burden [8]. However, despite traditional treatment of HF, mortality remains very high (20% within the first year of diagnosis and around half of patients after 5 years) [3,9]. In addition, HF is the main cause of hospitalization in Europe and the United States in subjects ≥65 years [2,10], and it is expected that the number of HF hospitalizations will increase by 50% in the following 25 years [2,11]. In Spain, more than 25% of cardiac hospitalizations are due to HF [12]. Fortunately, in the last few years, a number of clinical trials have shown a significant reduction in the risk of cardiovascular mortality and HF hospitalizations in both HF with reduced left ventricular ejection fraction (HFrEF), with sacubitril/valsartan and some sodium-glucose co-transporter-2 inhibitors (SGLT2i) [13,14,15,16], and HF with preserved left ventricular ejection fraction (HFpEF), with some SGLT2i, such as empagliflozin or dapagliflozin [17,18]. 

However, few contemporary studies have estimated the incidence and prevalence of HF stratified by EF (HFrEF, HFpEF, and HF with mildly reduced EF-HFmrEF-) or are limited to some regions. Additionally, recent data about the clinical profile and management of patients with HF stratified by EF are scarce [19,20,21,22,23,24,25,26,27]. As a result, to optimize the current approach of these patients, new data are warranted. This study aimed to address the knowledge gap in Spain regarding the epidemiology of HF by EF status by estimating the incidence, prevalence, patient characteristics, treatment, and outcomes through the analysis of a nationally representative Spanish database. 

## 2. Methods

This was a retrospective and observational cohort study using data from the BIG-PAC database in Spain. The BIG-PAC dissociated database is a nationally representative, longitudinal database combining healthcare data from various primary care and hospital centers across seven Spanish Autonomous Communities. Data on 1.8 million patients are available from 2012 onwards and are updated every month. Information includes demographics, clinical information, drug prescriptions, laboratory tests, and mortality. Many studies have confirmed the representativeness of the Spanish population and its ability to determine the clinical profile and management of outcomes [7,8,28]. This study was approved by the Investigation Ethics Committee of Consorci Sanitari from Terrassa. As this was a secondary data study and data were fully anonymized and dissociated from patients, no informed consent was required. 

All adults with at least one year of enrollment in the database (from 1 January 2013 to 30 September 2019) were included in the incidence and prevalence estimations. Two prevalent HF cohorts were defined at two index dates: 1 January 2016 and 1 January 2019. These cohorts included patients with at least one inpatient or outpatient HF diagnosis, at least one year of continuous enrollment before the corresponding index date, and at least 18 years of age at the index date. These cohorts were used to determine changes in the clinical profile and management during the period between the two index dates. Patients were excluded if they had less than one year of continuous enrollment before the index date, <18 years at the index date, or had chronic kidney disease stage V that required dialysis at any time before the index date (patients could develop CKD stage V requiring dialysis during the follow-up).

Baseline characteristics and treatments were determined for both prevalent cohorts. One-year event rates were assessed in the 2016 cohort (but not in the 2019 cohort, as data were available only until September 2019). Outcomes were defined according to the ICD-10. The data were stratified by EF subgroups: HFpEF: EF ≥ 50%; HFrEF: EF ≤ 40%; HFmrEF: EF > 40- < 50%; HF with unspecified EF (HFuEF): patients without an echocardiographic result in the data.

In both prevalent HF cohorts, baseline data, including demographics, HF data, cardiovascular risk factors, vascular disease, chronic kidney disease by stage [29], other comorbidities, and the Charlson Comorbidity Index [30] were collected. Comorbidities were based on data at any time up to the index date, unless otherwise specified. The International Classification of Diseases (ICD)-9 and ICD-10 codes (https://eciemaps.mscbs.gob.es (accessed on 9 March 2022)) were considered for the diagnosis of comorbidities (Appendix A).

Treatments one year before the index date were recorded from the registries for dispensing medicines according to the Anatomical Therapeutic Chemical Classification System (Appendix A) [31]. HF treatments included angiotensin-converting enzyme inhibitors (ACEi) or angiotensin receptor blockers (ARB), dual angiotensin receptor and neprilysin inhibition (ARNI), beta blockers, aldosterone antagonists, digoxin, ivabradine, hydralazine, and diuretics (excluding aldosterone antagonists). Other cardiovascular drugs and antidiabetic drugs were also recorded. 

### Statistical Analysis

No formal sample size calculation was conducted. All patients in the BIG-PAC database meeting the inclusion criteria and with no exclusion criteria were included in the study. The event rate for HF (per person time) was calculated by dividing the number of patients with incident HF (new HF inpatient/outpatient diagnoses) during the study period by the total person time contributed by all adult patients in the database without prevalent HF. The event rate for HF was reported per 100 person-years. Annual prevalence (per calendar year) was estimated as all patients with a qualifying HF diagnosis at the beginning of a calendar year divided by all adult patients alive and enrolled at the beginning of that calendar year and who had been continuously enrolled during the entire year prior. Prevalence and incidence were reported overall and for each year during the study period (all patients were included for prevalence and incidence estimation). Baseline characteristics and treatments were summarized using descriptive statistics and stratified by EF subgroups. The qualitative variables were described by their absolute and relative frequency distributions. Quantitative variables were described by measures of central tendency (mean, median) and dispersion (standard deviation, interquartile range). Annualized event rates for myocardial infarction, stroke, and HF hospitalization were calculated overall and in each EF subgroup. Event rates were calculated as the total number of events of interest divided by the total person time of follow-up from the index date (Poisson exact 95% confidence intervals [CI] were calculated). Patients were followed from the index date until death, loss to follow-up, or study end date. Patients were included regardless of their prior history of stroke or myocardial infarction, and were not censored at the first occurrence. Event rates were reported per 1000 person-years with 95% CI. The results in the HFpEF, HFmrEF, and HFuEF subgroups were compared with the HFrEF subgroup. To compare continuous variables between EF subtypes, a two-sample t-test was used for normally distributed variables and the Mann–Whitney U test for non-normally distributed variables. The chi-square test was used for categorical variables. A level of statistical significance of 0.05 was applied in all statistical tests. The data were analyzed using the statistical package SPSS v25.0 (SPSS Inc., Chicago, IL, USA).

## 3. Results

Event rates of HF are shown in Figure 1 and Appendix A. Overall, the HF event rate was 0.32 per 100 person-years but increased from 0.27 in 2013 to 0.37 per 100 person-years in 2018 (0.35 per 100 person-years in 2019 until September). Overall, these numbers were 0.14, 0.09, and 0.02 per 100 person-years for patients with HFrEF, HFpEF, and HFmrEF, respectively. In all cases, the incidence of HF increased from 2013 to 2018.

The overall prevalence of HF was 2.34% but increased from 2.07% to 2.44%, respectively (2.37% in 2019 until September). Overall, these numbers were 1.12%, 0.91%, and 0.10% for patients with HFrEF, HFpEF, and HFmrEF, respectively. In all cases, there was a trend toward an increase in the prevalence of HF over the years evaluated (Figure 2 and Appendix A).

Baseline characteristics and treatments in the prevalent 2016 HF cohort are presented in Table 1. Overall, the mean age was 78.8 ± 11.8 years, 53.0% were men, 42.0% were on New York Heart Association (NYHA) functional class II, and 41.0% were on NYHA functional class III. With regard to HF treatments, 74.1% were taking diuretics, 67.3% ACEi/ARB, 61.2% beta blockers, 23.4% aldosterone antagonists, and 5.2% SGLT2i. With regard to the type of HF, 48.5% had HFrEF, 38.6% had HFpEF, and 4.2% had HFmrEF (with the rest being of unknown EF). Compared with patients with HFrEF, patients with HFmrEF were older, more commonly women, and had more hypertension, dyslipidemia, and atrial fibrillation, but less diabetes, coronary artery disease, peripheral artery disease, and chronic obstructive pulmonary disease. Regarding HF treatments, patients with HFmrEF took fewer diuretics, ACEi/ARB, SGLT2i, digoxin, and ivabradine. Compared to those patients with HFrEF, patients with HFpEF were older, more commonly women, more patients were on NYHA functional class II, and had more dyslipidemia and atrial fibrillation, but less diabetes, coronary artery disease, CKD, stroke, peripheral artery disease, chronic obstructive pulmonary disease, and dementia. All HF treatments were more commonly prescribed in patients with HFrEF than in patients with HFpEF. The clinical profile and management were similar in those patients with HFpEF, regardless EF (50 to <60% vs. ≥60%).

Baseline characteristics and treatments in the prevalent 2019 HF cohort are presented in Table 2. The clinical profile of this population was very close to that of the 2016 cohort, including differences according to the type of HF. However, in the 2019 cohort, more patients were taking the recommended HF drugs. Thus, 69.8% were taking diuretics, 67.8% ACEi/ARB, 65.9% beta blockers, 27.7% aldosterone antagonists, 12.0% ARNI, and 5.1% SGLT2i. HF treatments were more commonly prescribed in patients with HFrEF versus HFmrEF or HFpEF.

Event rates of HF hospitalization in the prevalent 2016 HF cohort were 231.6 per 1000 person-years, respectively. The rates of HF hospitalization were significantly higher among patients with HFrEF vs. HFmrEF or HFpEF (Figure 3). Outcomes observed in the 2 subgroups of patients with HFpEF (EF 50 to <60% vs. ≥60%) were comparable (Appendix A). 

## 4. Discussion

Our study showed in a wide sample of subjects, representative of the Spanish population, that the overall HF incidence and prevalence were 0.32 per 100 person-years and 2.34%, respectively, but increased over time. Nearly half of the patients had HFrEF, approximately 40% had HFpEF, and 4% had HFmrEF. Although there were relevant differences in the clinical profile and management of patients according to the type of HF, a significant proportion of patients were not still taking the recommended HF drugs. 

Overall, it has been estimated that in Europe, the incidence of HF is around 0.5/100 person-years, and in the United States, it is approximately 0.6–0.79/100 person-years, after 45 years of age and 2.1/100 person-years, after 65 years of age [1,2]. The HF prevalence seems to be 1–2% of the adult population [1]. However, the prevalence of HF is not homogeneous, as it increases with age, from 1% in individuals under 55 years to more than 10% in subjects 70 years or older [1,5]. In our study, the event rate was 0.32/100 person-years, but increased every year, up to 0.37/100 person-years in 2018. Similarly, the overall prevalence of HF was 2.34% but increased every year, up to 2.44% in 2018. Although disparities in the clinical profiles of patients with HF across countries could explain these differences, it has also been reported that HF is commonly underdiagnosed, and more efforts should be performed to facilitate the early identification of these patients [32,33]. Our data strongly suggest that, in recent years, there has been an increase in both the incidence and prevalence of HF, regardless of the type of HF. It is likely that the aging of the population and an improved diagnosis of HF during this period could have an impact on the increased incidence and prevalence of HF [34].

In our study (mean age 79 years), approximately 49% of patients had HFrEF, 38% had HFpEF, 4% had HFmrEF, and 9% had HFuEF (54%, 42%, and 4%, respectively, if HFuEF was excluded). In a prospective observational study including 3580 newly diagnosed patients with HF (mean age 68 years) from university hospitals in Catalonia, Spain, the proportion of patients with HFrEF, HFpEF, and HFmrEF was 62%, 24%, and 14%, respectively [19]. In another study that included 1420 patients hospitalized for acute HF (mean age 72 years) from 20 Spanish hospitals, 41% of patients had HFrEF, 43% had HFpEF, and 16% had HFmrEF [20]. Data from 1061 Japanese patients (median age 81 years) showed that 61% had HFpEF and 39% had HFrEF [24]. In another study that analyzed 2601 hospitalized HF patients (mean age 64 years), 62% had HFrEF, 25% had HFpEF, and 13% had HFmrEF [25]. Although some studies have reported that half of all patients with HF have HFpEF, all these data indicate that the prevalence of HFpEF is markedly associated with age. Thus, as age increases, the proportion of patients with HFpEF rises [35]. 

Our study showed that there were relevant differences in the clinical profiles of patients with HFrEF or HFpEF. Thus, patients with HFpEF were older, more commonly women, and had more atrial fibrillation. In contrast, patients with HFrEF had more atherosclerotic cardiovascular disease. Other studies have some similar differences in the clinical profile, except for hypertension, which is usually more common among patients with HFpEF [19,20,24,25,36]. In fact, ischemic heart disease and hypertension are the most common causes of HFrEF and HFpEF, respectively [1]. Nevertheless, comorbidities are very common in both conditions, and only through a comprehensive approach can the maximum benefit be obtained [37]. In addition, we analyzed the group of patients with HFpEF in 2 subgroups (EF 50 to <60% vs. ≥60%), as some clinical trials have suggested a different efficacy of some drugs according to EF [17,38]. However, the clinical profiles and management were similar in the whole HFpEF spectrum, regardless of EF, suggesting that it is a homogeneous population.

Patients with HFmrEF have intermediate characteristics between HFrEF and HFpEF patients. Thus, some studies have shown that whereas the etiology (high prevalence of ischemic heart disease) and clinical profile are closer to that of HFrEF, the clinical prognosis is closer to HFpEF, with a higher risk of non-cardiovascular adverse events when compared with HFrEF [39,40]. In our study, the clinical profile of patients with HFmrEF was closer to that of HFpEF than that of HFrEF (i.e., elderly patients, more commonly women, with more hypertension and atrial fibrillation but less ischemic heart disease).

Regarding HF treatments, in the 2016 prevalent HF cohort, among patients with HFrEF, around 76% of patients were taking ACEi/ARB, 64% beta blockers, 25% aldosterone antagonists, 22% digoxin, 9% ivabradine, 7% SGLT2i, and no patients were taking ARNI. However, in the 2019 prevalent HF cohort, these numbers were 76%, 70%, 37%, 21%, 7%, 7%, and 13%, respectively. Therefore, although there has been an improvement in the prescription of HF-recommended drugs, there is still much room for improvement. However, it should be taken into account that although current guidelines recommend as first-line therapy the use of ARNI (preferably)/ACEi, beta blockers, aldosterone antagonists, and SGLT2i [1], the PARADIGM-HF trial was published in 2014 [13] and the DAPA-HF, EMPEROR-Reduced, and SOLOIST-WHF trials were published in 2019, 2020, and 2021, respectively [14,15,16]. Remarkably, this slight improvement in the use of HF drugs during this period has translated into lower outcomes and healthcare costs [7]. Despite this, these figures are still low. Some reasons have emerged, trying to explain why HF medication is not always prescribed according to guidelines. First, underdiagnosis of HF. Although there has been an improvement in the diagnosis of HF in Spain in recent years, it remains an important problem in clinical practice. In fact, a proportion of patients are not diagnosed in the early stages, and some HF drugs are more difficult to prescribe later [33,34]. However, therapeutic inertia is the main factor that explains the relatively low use of HF medication. Thus, fear of adverse events in frailer patients or in advanced stages may reduce the use of these drugs or uptitration in some subjects. In addition, some physicians may think that symptoms are more related to medication than to HF itself. Low blood pressure or renal insufficiency can be a barrier to the prescription of HF medication use or uptitration. However, more careful management of these patients could reduce these potential obstacles. As a result, improving the transition of care, a higher use of HF clinics, the development of ambulatory disease management programs, enhancing the role of HF nurses, and a higher empowerment of patients and their families could be very helpful to reduce therapeutic inertia, leading to a better prescription of HF drugs [41]. Fortunately, our data showed that, despite patients in the 2019 cohort being older and having some more comorbidities than 2016 patients, HF treatment was more optimized.

On the other hand, although evidence about the benefits of HF drugs is less robust among patients with HFpEF and HFmrEF [1], our data show that they are clearly underused in clinical practice in these patients. Therefore, more efforts are required to reduce the gap between guidelines and clinical practice in the whole spectrum of HF patients [42,43].

In our study, outcomes were commonly related to an underuse of HF-recommended drugs. Although they were higher in HFrEF, rates of outcomes were also very significant among HFpEF patients and similar regardless of EF. In fact, the risk of developing complications is higher for HF patients of any ejection fraction than for some common cancers in both men and women [44]. European guidelines recommend for patients with HFpEF, the use of diuretics for symptom relief in those congested patients, and the screening and treatment of comorbidities [1]. After the publication of these guidelines, the EMPEROR-Preserved and DELIVER trials have shown that empagliflozin and dapagliflozin reduce the risk of primary outcomes among this population [17,18]. Additionally, different meta-analyses have reported that SGLT2i reduces cardiovascular death and HF hospitalization among patients with HF, regardless of HF status [45,46]. The recent 2022 AHA/ACC/HFSA guidelines recommend strict blood pressure control for HFpEF patients. In addition, these guidelines consider that SGLT2i can be beneficial in decreasing HF hospitalizations and cardiovascular mortality, and in selected patients, the use of aldosterone antagonists, ARB or ARNI should be considered to decrease HF hospitalizations, particularly among patients with HF and EF on the lower end of this spectrum [47]. In our study, in the 2019 prevalent HF cohort, 59% of patients with HFpEF were taking ACEi/ARB, 61% beta blockers, 18% aldosterone antagonists, 11% ARNI, and 3% SGLT2i. Therefore, more effort is needed to improve the management of these patients, particularly with the higher use of SGLT2i.

Our study showed that among patients with HF, the risk of outcomes, particularly HF hospitalization, was high in all EF groups. These results are in line with previous studies [2,6,7,8,9,10,11]. Therefore, as HF treatment has been associated with substantial reductions in HF hospitalizations, to actually reduce the HF burden, the prescription of drugs that have demonstrated clinical benefit in this setting is mandatory [1,47].

Our study has some limitations. This was an observational cohort study that used secondary data from electronic health records. Therefore, only data that were recorded in the electronic clinical history could be collected, leading to a possible underdiagnosis of some variables (i.e., tachycardiomyopathy due to AF, cardiotoxicity from cancer treatment, etc.). In fact, only those patients with an HF diagnosis (ICD-10 code) at any time before the index date could be included in the study. However, this is the best design to actually represent clinical practice, as no specific intervention was performed to be included in the study. In addition, the high number of HF patients included, and the robustness of the data may reduce potential bias. Moreover, to our knowledge, this is the first study in Spain assessing the epidemiology and burden of HF, with a particular focus on EF subgroups in a nationally representative HF population. However, despite the fact that in Spain, the public health system allows free access to the entire population, the organization of care for patients with HF may vary between regions (e.g., availability of HF units, HF nurses, etc.), and this could have an impact on the different prescriptions of HF drugs and the risk of outcomes. On the other hand, event rates were only provided for the 2016 cohort, as data were recorded only until September 2019. In addition, data on mortality were available only in the incident cohort but not in the prevalent cohorts.

In conclusion, in this representative study of the Spanish population, more than 2% of patients had HF, of which nearly half had HFrEF, and 40% had HFpEF. However, the prevalence of HF has increased over time. In addition, the clinical profile of patients varies according to the type of HF. Finally, the risk of outcomes, particularly HF hospitalization, is high, and this could be partially related to an insufficient intensification of HF treatment. Therefore, more effort is required to optimize the management of these patients.

## Figures and Tables

**Figure 1 jcm-11-05199-f001:**
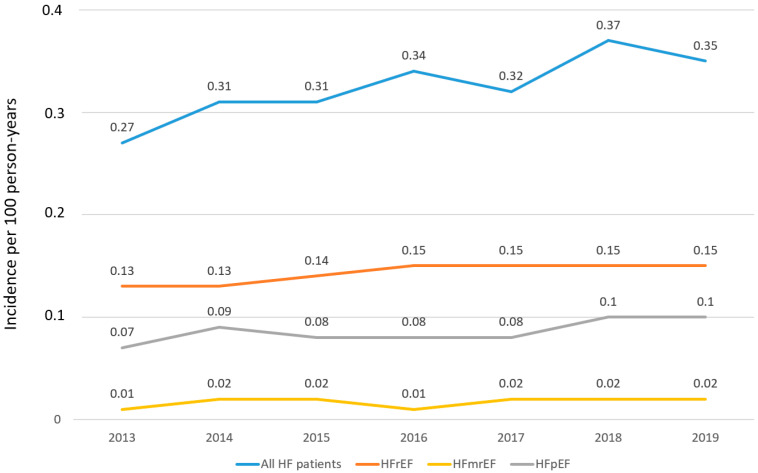
Incidence rates of HF. Abbreviations: HF = Heart failure; HFmrEF = Heart failure with mildly reduced ejection fraction; HfpEF = Heart Failure with preserved ejection fraction; HFrEF = Heart failure with reduced ejection fraction; HfuEF = Heart Failure with un-specified ejection fraction. Data were recorded until September 2019.

**Figure 2 jcm-11-05199-f002:**
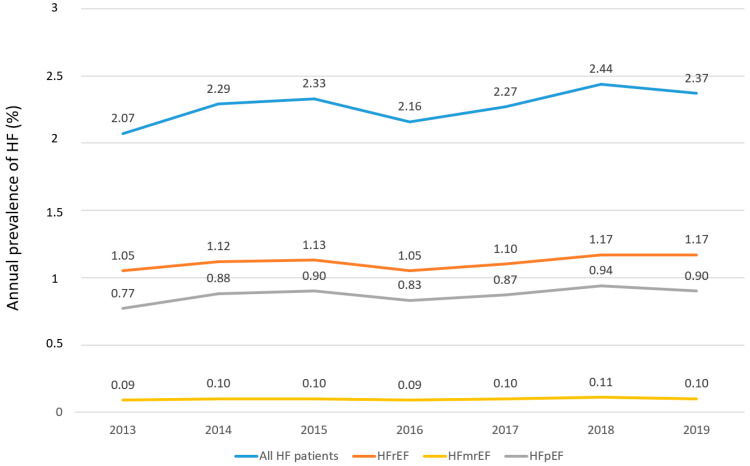
Annual prevalence of HF. Abbreviations: HF = Heart failure; HFmrEF = Heart failure with mildly reduced ejection fraction; HFpEF = Heart Failure with preserved ejection fraction; HFrEF = Heart failure with reduced ejection fraction; HFuEF = Heart Failure with un-specified ejection fraction. Data were recorded until September 2019.

**Figure 3 jcm-11-05199-f003:**
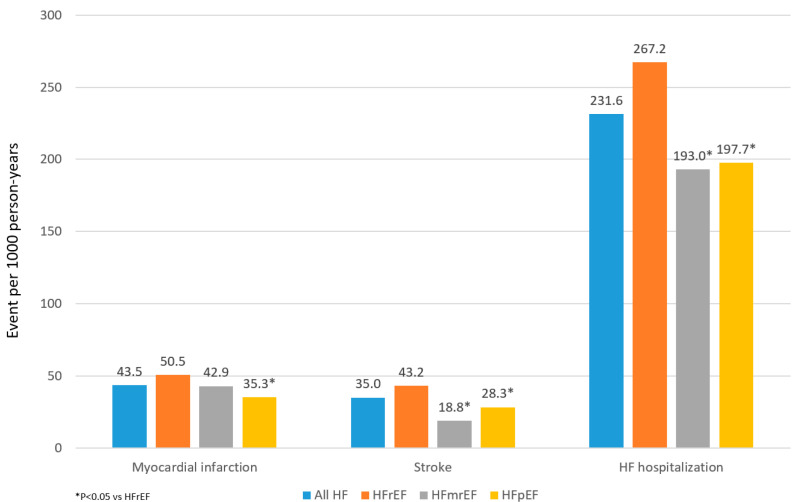
Event rates in the 2016 HF cohort. Abbreviations: HF = Heart failure; HFmrEF = Heart failure with mildly reduced ejection fraction; HFpEF = Heart Failure with preserved ejection fraction; HFrEF = Heart failure with reduced ejection fraction; HFuEF = Heart Failure with un-specified ejection fraction.

**Table 1 jcm-11-05199-t001:** Baseline characteristics and treatments in the prevalent heart failure cohort (index date 1 January 2016).

	HF Prevalent Cohort (n = 21,297; 100%)	HFrEF (n = 10,323; 48.5%)	HFmrEF (n = 903; 4.2%)	HFpEF (n = 8225; 38.6%)	HFpEF (50 to <60) (n = 2995; 14.1%)	HFpEF (≥60) (n = 5230; 24.6%)	HFuEF (n = 1846; 8.7%)	*p*-Value (HFmrEF vs. HFrEF)	*p*-Value (HFpEF vs. HFrEF)
**Biodemographic Data**									
Age (years) at index date Mean (SD) Median (25th75th percentile) Range (min–max)	78.8 (11.8) 79.1 (71.9–87.6) (18–102.8)	73.6 (9.7) 73.8 (67.5–79.8) (18–96.36)	81.7 (9.9) 80.6 (75.6–89.4) (21.6–98.8)	84.0 (11.4) 85.6 (78.2–91.9) (18.5–102.8)	84.1 (11.3) 85.7 (78.4–89.4) (18.6–98.8)	84.0 (11.4) 85.5 (78.2–91.9) (18.5–100.5)	83.3 (11.9) 85.5 (78.0–91.9) (18.9–100.5)	<0.001	<0.001
Age groupsn (%) <45 years 45–64 years 65–74 years 75–84 years ≥85 years	251 (1.2) 2007 (9.4) 5124 (24.1) 6981 (32.8) 6934 (32.6)	122 (1.2) 1479 (14.3) 4087 (39.6) 3314 (32.1) 1321 (12.8)	2 (0.2) 34 (3.8) 168 (18.6) 346 (38.3) 353 (39.1)	103 (1.3) 376 (4.6) 695 (8.5) 2754 (33.5) 4297 (52.2)	37 (1.2) 134 (4.5) 244 (8.2) 1000 (33.4) 1580 (52.8)	66 (1.3) 242 (4.6) 451 (8.6) 1754 (33.5) 2717 (52.0)	24 (1.3) 118 (6.4) 174 (9.4) 567 (30.7) 963 (52.2)		
Gendermalen (%)	11,278 (53.0)	6782 (65.7)	440 (48.7)	3068 (37.3)	1102 (36.8)	1966 (37.6)	988 (53.5)	<0.001	<0.001
NYHA at index daten (%) Class I Class II Class III Class IV Unknown	2137 (10.0) 8949 (42.0) 8728 (41.0) 1013 (4.8) 470 (2.2)	1030 (10.0) 3689 (35.7) 4750 (46.0) 612 (5.9) 242 (2.3)	91 (10.1) 332 (36.8) 411 (45.5) 56 (6.2) 13 (1.4)	817 (9.9) 4176 (50.8) 2783 (33.8) 280 (3.4) 169 (2.1)	322 (10.8) 1504 (50.2) 1016 (33.9) 92 (3.1) 61 (2.0)	495 (9.5) 2672 (51.1) 1767 (33.8) 188 (3.6) 108 (2.1)	199 (10.8) 752 (40.7) 784 (42.5) 65 (3.5) 46 (2.5)	0.494	<0.001
Cardiovascular risk factorsn (%)									
Hypertension	14,379 (67.5)	6885 (66.7)	662 (73.3)	5550 (67.5)	2021 (67.5)	3529 (67.5)	1282 (69.5)	<0.001	0.261
Dyslipidemia	10,457 (49.1)	4681 (45.4)	457 (50.6)	4384 (53.3)	1601 (53.5)	2783 (53.2)	935 (50.7)	0.002	<0.001
Diabetes type 1	844 (4.0)	499 (4.8)	32 (3.5)	258 (3.1)	100 (3.3)	158 (3.0)	55 (3.0)	0.080	<0.001
Diabetes type 2	6772 (31.8)	3331 (32.3)	236 (26.1)	2630 (32.0)	1127 (37.6)	1503 (28.7)	575 (31.2)	<0.001	0.672
Vascular diseasen (%)									
Coronary artery disease	8124 (38.2)	4520 (43.8)	288 (31.9)	2653 (32.3)	971 (32.4)	1682 (32.2)	663 (35.9)	<0.001	<0.001
Chronic kidney disease Stage Unknown Stage I Stage II Stage III Stage IV Stage V	6452 (30.3) 2706 (12.7) 179 (0.8) 644 (3.0) 2225 (10.5) 524 (2.5) 174 (1.0)	3411 (33.0) 1451 (14.1) 86 (0.8) 327 (3.2) 1179 (11.4) 281 (2.7) 87 (1.1)	272 (30.1) 106 (11.7) 6 (0.7) 19 (2.1) 106 (11.7) 26 (2.9) 9 (1.1)	2286 (27.8) 953 (11.6) 73 (0.9) 250 (3.0) 789 (9.6) 153 (1.9) 68 (0.9)	849 (28.4) 358 (12.0) 26 (0.9) 85 (2.8) 296 (9.9) 63 (2.1) 21 (0.9)	1437 (27.5) 595 (11.4) 47 (0.9) 165 (3.2) 493 (9.4) 90 (1.7) 47 (0.9)	483 (26.2) 196 (10.6) 14 (0.8) 48 (2.6) 151 (8.2) 64 (3.5) 10 (0.6)	0.073	<0.001
Myocardial Infarction	3174 (14.9)	1645 (15.9)	103 (11.4)	1110 (13.5)	384 (12.8)	726 (13.9)	316 (17.1)	<0.001	<0.001
Stroke	2254 (10.6)	1327 (12.9)	107 (11.9)	617 (7.5)	297 (9.9)	320 (6.1)	203 (11.0)	0.385	<0.001
Peripheral arterial disease	1074 (5.0)	616 (6.0)	24 (2.7)	337 (4.1)	146 (4.9)	191 (3.7)	97 (5.3)	<0.001	<0.001
Other comorbiditiesn (%)									
COPD	3319 (15.6)	1716 (16.6)	121 (13.4)	1202 (14.6)	441 (14.7)	761 (14.6)	280 (15.2)	0.012	<0.001
Atrial fibrillation	6785 (31.9)	2538 (24.6)	283 (31.3)	3364 (40.9)	1205 (40.2)	2159 (41.3)	600 (32.5)	<0.001	<0.001
Anemia within 1 year before index date	6540 (30.7)	3266 (31.6)	255 (28.2)	2503 (30.4)	910 (30.4)	1593 (30.5)	516 (28.0)	0.035	0.078
Cancer before index date	2776 (13.0)	1313 (12.72)	109 (12.1)	1077 (13.1)	368 (12.3)	709 (13.6)	277 (15.0)	0.574	0.449
Dementia	1058 (5.0)	568 (5.5)	45 (5.0)	360 (4.4)	168 (5.6)	192 (3.7)	85 (4.6)	0.510	<0.001
Charlson Comorbidity Index Mean (SD) Median (25th75th percentile)	3.0 (1.5) 3 (2–4)	3.1 (1.5) 3 (2–4)	3.0 (1.4) 3 (2–4)	3.0 (1.4) 3 (2–4)	3.3 (1.4) 3 (2–4)	2.8 (1.5) 2 (2–4)	3.0 (1.5) 3 (2–4)	0.066	<0.001
Medicationsn (%)									
HF drugs									
Diuretics	15,780 (74.1)	7759 (75.2)	649 (71.9)	5964 (72.5)	2174 (72.6)	3790 (72.5)	1408 (76.3)	0.029	<0.001
ACEi/ARB	14,335 (67.3)	7840 (76.0)	574 (63.6)	4806 (58.4)	1734 (57.9)	3072 (58.7)	1115 (60.4)	<0.001	<0.001
Beta-blockers	13,043 (61.2)	6631 (64.2)	602 (66.7)	4693 (57.1)	1711 (57.1)	2982 (57.0)	1117 (60.5)	0.143	<0.001
Aldosterone antagonists	4973 (23.4)	2609 (25.3)	207 (22.9)	1765 (21.5)	654 (21.8)	1111 (21.2)	392 (21.2)	0.118	<0.001
Digoxin	4311 (20.2)	2307 (22.4)	162 (17.9)	1437 (17.5)	526 (17.6)	911 (17.4)	405 (21.9)	0.002	<0.001
Ivabradine	1502 (7.1)	873 (8.5)	38 (4.2)	449 (5.5)	181 (6.0)	268 (5.1)	142 (7.7)	<0.001	<0.001
Hydralazine and nitrate	19 (0.1)	7 (0.1)	1 (0.1)	11 (0.1)	5 (0.2)	6 (0.1)	0	0.643	0.152
ARNI	0	0	0	0	0	0	0		
Other cardiovascular drugs at baseline									
Lipid-lowering drugs	11,282 (53.0)	5888 (57.0)	534 (59.1)	3925 (47.7)	1419 (47.4)	2506 (47.9)	935 (50.7)	0.222	<0.001
Any antiplatelet drugs ASA P2Y12 inhibitors DAPT (ASA + P2Y12)	7732 (36.3) 5269 (24.7) 2174 (10.2) 876 (4.1)	4223 (40.9) 3012 (29.2) 1181 (11.4) 442 (4.3)	294 (32.6) 200 (22.2) 80 (8.9) 30 (3.3)	2567 (31.2) 1647 (20.0) 708 (8.6) 346 (4.2)	921 (30.8) 596 (19.9) 264 (8.8) 144 (4.8)	1646 (31.5) 1051 (20.1) 444 (8.5) 202 (3.9)	648 (35.1) 410 (22.2) 205 (11.1) 58 (3.1)	<0.001 <0.001 0.019 0.168	<0.001 <0.001 <0.001 0.801
Anticoagulants	6048 (28.4)	2271 (22.0)	242 (26.8)	2987 (36.3)	1072 (35.8)	1915 (36.6)	548 (29.7)	0.001	<0.001
Calcium channel blockers	4461 (21.0)	751 (7.3)	154	3155 (38.4)	1153 (38.5)	2002 (38.3)	<0.001	(17.1)	<0.001
Nitrates	2428 (11.4)	1196 (11.6)	116 (12.9)	878 (10.7)	329 (11.0)	549 (10.5)	238 (12.9)	0.258	0.050
Nicorandil	14 (0.1)	5 (0.05)	0	7 (0.1)	3 (0.10)	4 (0.1)	2 (0.1)	0.508	0.329
Antidiabetic drugs at baseline									
Metformin	6414 (30.1)	3428 (33.2)	283 (31.3)	2138 (26.0)	763 (25.5)	1375 (26.3)	565 (30.6)	0.253	<0.001
Sulfonylurea	2626 (12.3)	1339 (13.0)	141 (15.6)	918 (11.2)	321 (10.7)	597 (11.4)	228 (12.4)	0.024	<0.001
DPP-4i	2509 (11.8)	1466 (14.2)	67 (7.4)	763 (9.3)	277 (9.3)	486 (9.3)	213 (11.5)	<0.001	<0.001
Insulin	1571 (7.4)	790 (7.7)	58 (6.4)	591 (7.2)	196 (6.5)	395 (7.6)	132 (7.2)	0.180	0.228
SGLT2i	1115 (5.2)	704 (6.8)	34 (3.8)	267 (3.3)	89 (3.0)	178 (3.4)	110 (6.0)	<0.001	<0.001
Other glucose-lowering drugs	822 (3.9)	476 (4.6)	38 (4.2)	230 (2.8)	86 (2.9)	144 (2.8)	78 (4.2)	0.579	<0.001
GLP1-RA	229 (1.1)	98 (1.0)	13 (1.4)	98 (1.2)	35 (1.2)	63 (1.2)	20 (1.1)	0.153	0.109
Other drugs at baseline									
PPIs	13,942 (65.5)	7704 (74.6)	486 (53.8)	4608 (56.0)	1655 (55.3)	2953 (56.5)	1144 (62.0)	<0.001	<0.001
NSAIDs	9978 (46.9)	5410 (52.4)	297 (32.9)	3399 (41.3)	1215 (40.6)	2184 (41.8)	872 (47.2)	<0.001	<0.001
Number of drugs at index date									
0	5 (0.02)	1 (0.01)	0	3 (0.04)	3 (0.1)	0	1 (0.05)		
1	88 (0.4)	22 (0.2)	5 (0.6)	55 (0.7)	18 (0.6)	37 (0.7)	6 (0.3)		
2	460 (2.2)	127 (1.2)	38 (4,2)	261 (3.2)	97 (3,2)	164 (3,1)	34 (1.8)		
3	1489 (7.0)	508 (4,9)	86 (9,5)	751 (9.1)	258 (8.6)	493 (9,4)	144 (7.8)		
4	2884 (13.5)	1134 (11.0)	172 (19,1)	1310 (15.9)	516 (17.2)	794 (15,2)	268 (14.5)		
5	4253 (20.0)	1865 (18,1)	185 (20,5)	1825 (22.2)	658 (22.0)	1167 (22,3)	378 (20.5)		
≥6	12,118 (57.0)	6666 (64.6)	417 (46.2)	4020 (48.9)	1445 (48.2)	2575 (49.2)	1015 (55.0)		

All treatments were assessed within 12 months before index. Patients on combination drugs were counted in each respective treatment class. Therefore, each treatment class included patients undergoing monotherapy and combination therapy. Anemia is expected to be underreported, as it can be a symptom rather than a diagnosis. The lookback period for all comorbidities was any time before the index date (event date < index date), unless otherwise specified; the lookback period for all prescriptions was 12 months prior to the index date. Abbreviations: ACE = Angiotensin-converting enzyme; ARB = angiotensin receptor II blocker; ARNI = Dual angiotensin receptor and neprilysin inhibition; ASA = Acetylsalicylic acid; COPD = Chronic obstructive pulmonary disease; DAPT = dual antiplatelet therapy; DPP4i = Dipeptidyl peptidase 4 inhibitors; GLP1-RA = Glucagon like peptide 1 receptor agonist; HF = Heart failure; HFmrEF = Heart failure with mildly reduced ejection fraction; HFpEF = Heart Failure with preserved ejection fraction; HFrEF = Heart Failure with reduced ejection fraction; HFuEF = Heart Failure with unspecified ejection fraction; NSAIDs = Nonsteroidal anti-inflammatory drugs; NYHA= New York Heart Association; PPI = Proton pump inhibitors; P2Y12 inhibitors = adenosine diphosphate (ADP) receptor antagonists; SD: standard deviation; SGLT2i = Sodium-glucose co-transporter-2 inhibitors.

**Table 2 jcm-11-05199-t002:** Baseline characteristics and treatments in the prevalent heart failure cohort (index date 1 January 2019).

	HF Prevalent Cohort (n = 23,806; 100%)	HFrEF (n = 11,746; 49.3%)	HFmrEF (n = 1031; 4.3%)	HFpEF (n = 9062; 38.1%)	HFpEF (50 to <60) (n = 3321; 14.0%)	HFpEF (≥60) (n = 5741; 24.1%)	HFuEF (n = 1967; 8.3%)	*p*-Value (HFmrEF vs. HFrEF)	*p*-Value (HFpEF vs. HFrEF)
Biodemographic data									
Age (years) at index date Mean (SD) Median (25th75th percentile) Range (min–max)	78.8 (12.4) 79.3 (70.9–88.4) (18.0–99.5)	73.7 (10.1) 73.2 (68.3–80.2) (18.0–99.4)	81.0 (11.2) 81.0 (75.9–89.3) (18.1–99.3)	84.4 (12.1) 86.9 (79.0–93.1) (18.9–99.5)	84.3 (12.3) 86.9 (79.0–89.3) (19.7–99.3)	84.5 (12.1) 87.0 (79.1–93.1) (18.9–99.4)	82.4 (13.8) 86.7 (73.2–93.1) (19.5–99.4)	<0.001	<0.001
Age groupsn (%) <45 years 45–64 years 65–74 years 75–84 years ≥85 years	147 (0.6) 1918 (8.1) 5995 (25.2) 8063 (33.9) 7683 (32.3)	52 (0.4) 1051 (9.0) 5296 (45.1) 4320 (36.8) 1027 (8.7)	6 (0.6) 57 (5.5) 129 (12.5) 485 (47.0) 354 (34.3)	64 (0.7) 663 (7.3) 214 (2.4) 2918 (32.2) 5203 (57.4)	26 (0.9) 242 (7.3) 80 (2.4) 1076 (32.4) 1897 (57.1)	38 (0.7) 421 (7.3) 134 (2.3) 1842 (32.1) 3306 (57.6)	25 (1.3) 147 (7.5) 356 (18.1) 340 (17.3) 1099 (55.9)		
Gendermalen (%)	12,780 (53.7)	7713 (65.7)	523 (50.7)	3528 (38.9)	1268 (38.2)	2260 (39.4)	1016 (51.7)	<0.001	<0.001
NYHA at index daten (%) Class I Class II Class III Class IV Unknown	2391 (10.0) 9967 (41.9) 9785 (41.1) 1132 (4.8) 531 (2.2)	1194 (10.2) 4176 (35.6) 5400 (46.0) 701 (6.0) 275 (2.3)	99 (9.6) 367 (35.6) 489 (47.4) 58 (5.6) 18 (1.8)	879 (9.7) 4627 (51.1) 3079 (34.0) 293 (3.2) 184 (2.0)	312 (9.4) 1669 (50.3) 1153 (34.7) 117 (3.5) 70 (2.1)	567 (9.9) 2958 (51.5) 1926 (33.6) 176 (3.1) 114 (2.0)	219 (11.1) 797 (40.5) 817 (41.5) 80 (4.1) 54 (2.8)	0.665	<0.001
Cardiovascular risk factorsn (%)									
Hypertension	16,168 (67.9)	7897 (67.2)	775 (75.2)	6133 (67.7)	2229 (67.1)	3904 (68.0)	1363 (69.3)	<0.001	0.495
Dyslipidemia	11,650 (48.9)	5373 (45.7)	492 (47.7)	4800 (53.0)	1727 (52.0)	3073 (53.5)	985 (50.1)	0.222	<0.001
Diabetes type 1	960 (4.0)	565 (4.8)	41 (4.0)	279 (3.1)	128 (3.9)	151 (2.6)	75 (3.8)	0.227	<0.001
Diabetes type 2	7603 (31.9)	3794 (32.3)	287 (27.8)	2899 (32.0)	1288 (38.8)	1611 (28.1)	623 (31.7)	0.003	0.635
Vascular diseasen (%)									
Coronary artery disease	8986 (37.8)	5053 (43.0)	324 (31.4)	2871 (31.7)	1071 (32.3)	1800 (31.4)	738 (37.5)	<0.001	<0.001
Chronic kidney disease Stage Unknown Stage I Stage II Stage III Stage IV Stage V	7271 (30.5) 2985 (12.5) 190 (0.8) 725 (3.01) 2586 (10.9) 576 (2.4) 209 (1.0)	3935 (33.5) 1642 (14.0) 99 (0.8) 376 (3.2) 1390 (11.8) 316 (2.7) 112 (1.1)	328 (31.8) 132 (12.8) 8 (0.8) 34 (3.3) 122 (11.8) 21 (2.0) 11 (1.1)	2490 (27.5) 1001 (11.1) 72 (0.8) 269 (3.0) 896 (9.9) 183 (2.0) 69 (0.9)	923 (27.8) 366 (11.0) 19 (0.6) 97 (2.9) 344 (10.4) 75 (2.3) 22 (0.9)	1567 (27.3) 635 (11.1) 53 (0.9) 172 (3.0) 552 (9.6) 108 (1.9) 47 (0.9)	518 (26.3) 210 (10.7) 11 (0.6) 46 (2.3) 178 (9.1) 56 (2.9) 17 (0.6)	0.271	<0.001
Myocardial Infarction	3571 (15.0)	1904 (16.2)	118 (11.5)	1239 (13.7)	458 (13.8)	781 (13.6)	310 (15.8)	<0.001	<0.001
Stroke	2525 (10.6)	1497 (12.7)	122 (11.8)	692 (7.6)	320 (9.6)	372 (6.5)	214 (10.9)	0.399	<0.001
Peripheral arterial disease	1169 (4.9)	684 (5.8)	29 (2.8)	349 (3.9)	164 (4.9)	185 (3.2)	107 (5.4)	<0.001	<0.001
Other comorbiditiesn (%)									
COPD	3658 (15.4)	1905 (16.2)	127 (12.3)	1319 (14.6)	503 (15.2)	816 (14.2)	307 (15.6)	0.001	0.001
Atrial fibrillation	7596 (31.9)	2931 (25.0)	315 (30.6)	3718 (41.0)	1353 (40.7)	2365 (41.2)	632 (32.1)	<0.001	<0.001
Anemia within 1 year before index date	7276 (30.6)	3721 (31.7)	290 (28.1)	2707 (29.9)	973 (29.3)	1734 (30.2)	558 (28.4)	0.018	0.005
Cancer before index date	3160 (13.3)	1514 (12.9)	125 (12.1)	1230 (13.6)	435 (13.1)	795 (13.9)	291 (14.8)	0.481	0.148
Dementia	1264 (5.31)	655 (5.6)	64 (6.2)	440 (4.9)	200 (6.0)	240 (4.2)	105 (5.3)	0.399	0.021
Charlson Comorbidity Index Mean (SD) Median (25th75th percentile)	3.1 (1.4) 3 (2–4)	3.2 (1.5) 3 (2–4)	3.1 (1.4) 3 (2–4)	3.1 (1.4) 3 (2–4)	3.5 (1.4) 3 (2–4)	2.8 (1.3) 3 (2–4)	3.1 (1.3) 3 (2–4)	0.455 3 (2–4)	<0.001
Medicationsn (%)									
HF drugs									
Diuretics	16,607 (69.8)	8262 (70.3)	689 (66.8)	6277 (69.3)	2309 (69.5)	3968 (69.1)	1379 (70.1)	0.018	0.095
ACEi/ARB	16,134 (67.8)	8957 (76.3)	631 (61.2)	5331 (58.8)	2017 (60.7)	3314 (57.7)	1215 (61.8)	<0.001	<0.001
Beta-blockers	15,690 (65.9)	8164 (69.5)	694 (67.3)	5537 (61.1)	1996 (60.1)	3541 (61.7)	1295 (65.8)	0.143	<0.001
Aldosterone antagonists	6594 (27.7)	4324 (36.8)	185 (17.9)	1584 (17.5)	612 (18.4)	972 (16.9)	501 (25.5)	0.126	0.020
Digoxin	4542 (19.1)	2481 (21.1)	172 (16.7)	1500 (16.6)	571 (17.2)	929 (16.2)	389 (19.8)	0.001	<0.001
ARNI	2857 (12.0)	1468 (12.5)	112 (10.9)	1037 (11.4)	405 (12.2)	632 (11.0)	240 (12.2)	<0.001	<0.001
Ivabradine	1440 (6.1)	848 (7.2)	40 (3.9)	424 (4.7)	161 (4.9)	263 (4.6)	128 (6.59)	<0.001	<0.001
Hydralazine and nitrate	0	0	0	0	0	0	0		
Other cardiovascular drugs at baseline									
Lipid-lowering drugs	12,700 (53.4)	6703 (57.1)	606 (58.8)	4395 (48.5)	1611 (48.5)	2784 (48.5)	996 (50.6)	0.287	<0.001
Any antiplatelet drugs ASA P2Y12 inhibitors DAPT (ASA + P2Y12)	8710 (36.6) 5830 (24.5) 2417 (10.2) 953 (4.0)	4862 (41.4) 3349 (28.5) 1364 (11.6) 502 (4.3)	356 (34.5) 232 (22.5) 92 (8.9) 49 (4.8)	2819 (31.1) 1791 (19.8) 771 (8.5) 335 (3,7)	1017 (30.6) 643 (19.4) 281 (8.5) 118 (3.6)	1802 (31.4) 1148 (20.0) 490 (8.5) 217 (3.8)	673 (34.2) 458 (23.3) 190 (9.7) 67 (3.4)	<0.001 <0.001 0.009 0.468	<0.001 <0.001 <0.001 0.036
Anticoagulants	6640 (27.9)	2580 (22.0)	269 (26.1)	3239 (35,7)	1167 (35.1)	2072 (36.1)	552 (28.1)	0.002	<0.001
Calcium channel blockers	4815 (20.2)	833 (7.1)	157 (15.2)	3406 (37.6)	1197 (36.0)	2209 (38.5)	419 (21.3)	<0.001	<0.001
Nitrates	2710 (11.4)	1409 (12.0)	133 (12.9)	935 (10.3)	362 (10.9)	573 (10.0)	233 (11.9)	0.393	<0.001
Nicorandil	0	0	0	0	0	0	0		
Antidiabetic drugs at baseline									
Metformin	6936 (29.1)	3751 (31.9)	297 (28.8)	2323 (25.6)	824 (24.8)	1499 (26.1)	565 (28.7)	0.039	<0.001
Sulfonylurea	2874 (12.1)	1403 (11.9)	146 (14.2)	1094 (12.1)	417 (12.6)	677 (11.8)	231 (11.7)	0.037	0.778
DPP-4i	2785 (11.7)	1622 (13.8)	80 (7.8)	814 (9.0)	292 (8.8)	522 (9.1)	269 (13.7)	<0.001	<0.001
Insulin	1764 (7.4)	930 (7.9)	66 (6.4)	626 (6.9)	223 (6.7)	403 (7.0)	142 (7.2)	0.082	0.006
SGLT2i	1218 (5.1)	792 (6.7)	26 (2.5)	291 (3.2)	94 (2.8)	197 (3.4)	109 (5.5)	<0.001	<0.001
Other glucose-lowering drugs	948 (4.0)	552 (4.7)	37 (3.6)	270 (3.0)	117 (3.5)	153 (2.7)	89 (4.5)	0.103	<0.001
GLP1-RA	239 (1.0)	112 (1.0)	15 (1.5)	94 (1.0)	33 (1.0)	61 (1.1)	18 (0.9)	0.120	0.545
Other drugs at baseline									
PPIs	15,533 (65.3)	8724 (74.3)	540 (52.4)	5052 (55.8)	1864 (56.1)	3188 (55.5)	1217 (61.9)	<0.001	<0.001
NSAIDs	11,246 (47.2)	6217 (52.9)	344 (33.4)	3769 (41.6)	1386 (41.7)	2383 (41.5)	916 (46.6)	<0.001	<0.001
Number of drugs at index date									
0	11 (0.05)	3 (0.03)	1 (0.1)	7 (0.1)	6 (0.2)	1 (0.02)	0		
1	121 (0.5)	17 (0.1)	12 (1.2)	79 (0.9)	29 (0.9)	50 (0.9)	13 (0.7)		
2	532 (2.2)	129 (1.1)	37 (3.6)	321 (3.5)	118 (3.6)	203 (3.5)	45 (2.3)		
3	1693 (7.1)	511 (4.4)	105 (10.2)	947 (10.5)	361 (10.9)	586 (10.2)	130 (6.6)		
4	3193 (13.4)	1158 (9.9)	214 (20.8)	1548 (17.1)	586 (17.7)	962 (16.8)	273 (13.9)		
5	4720 (19.8)	2041 (17.4)	249 (24.2)	2012 (22.2)	736 (22.2)	1276 (22.2)	418 (21.3)		
≥6	13,536 (56.9)	7887 (67.1)	413 (40.1)	4148 (45.8)	1485 (44.7)	2663 (46.4)	1088 (55.3)		

All treatments were assessed within 12 months before index. Patients on combination drugs were counted in each respective treatment class. Therefore, each treatment class included patients undergoing monotherapy and combination therapy. Anemia is expected to be underreported, as it can be a symptom rather than a diagnosis. The lookback period for all comorbidities was any time before the index date (event date < index date), unless otherwise specified; the lookback period for all prescriptions was 12 months prior to the index date. Abbreviations: ACE = Angiotensin-converting enzyme; ARB = angiotensin receptor II blocker; ARNI = Dual angiotensin receptor and neprilysin inhibition; ASA = Acetylsalicylic acid; COPD = Chronic obstructive pulmonary disease; DAPT = dual antiplatelet therapy; DPP4i = Dipeptidyl peptidase 4 inhibitors; GLP1-RA = Glucagon like peptide 1 receptor agonist; HF = Heart failure; HFmrEF = Heart failure with mildly reduced ejection fraction; HFpEF = Heart Failure with preserved ejection fraction; HFrEF = Heart Failure with reduced ejection fraction; HFuEF = Heart Failure with unspecified ejection fraction; NSAIDs = Nonsteroidal anti-inflammatory drugs; NYHA= New York Heart Association; PPI = Proton pump inhibitors; P2Y12 inhibitors = adenosine diphosphate (ADP) receptor antagonists; SD: standard deviation; SGLT2i = Sodium-glucose co-transporter-2 inhibitors.

## Data Availability

This was a secondary data study using BIG PAC® database that can be obtained under appropriate request.

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
