# Peer review of "Prevalence, Characteristics, Management and Outcomes of Patients with Heart Failure with Preserved, Mildly Reduced, and Reduced Ejection Fraction in Spain"

_jcm, 2022, doi:10.3390/jcm11175199_

Round 1
Reviewer 1 Report
Escobar and colleagues have presented an impressive study with a huge patient population.
The methods were presented in a detailed and understandable way. The authors show new findings, the results are extensive and neatly elaborated.
The discussion could be longer in view of a very detailed results section. Also, the possible clinical relevance and potential therapeutic approach could be discussed more concretely.
For example:
What could be the specific reasons for an increase in incidence and prevalence?
What could be specific reasons that HF medication is not always prescribed according to guidelines?
Are there statistically significant differences between age groups in 2016 and 2019, as might be suspected from the data in the tables?
Has there possibly been a better diagnosis of HF for the Spanish population over the years?
Can we tell from the database, how often patients went to the doctor? This could possibly be calculated because of the monthly update. Can anything be said about access to medical care for the Spanish population? Are there differences between urban and rural populations?
The sentence in line 146/147 seems oddly worded. Is it really correct like that?
Overall Escobar and colleagues have presented a manuscript with data of interest for cardiologists, cardiac surgeons, other clinicians, epidemiologists, economists and scientists. It is an excellent work with important findings to understand HF and possible further therapeutic options.
Author Response
Escobar and colleagues have presented an impressive study with a huge patient population. The methods were presented in a detailed and understandable way. The authors show new findings, the results are extensive and neatly elaborated.
The discussion could be longer in view of a very detailed results section. Also, the possible clinical relevance and potential therapeutic approach could be discussed more concretely.
Thanks for your comments. The discussion has been expanded as required below.
For example:
What could be the specific reasons for an increase in incidence and prevalence?
Although this was not specifically analyzed in our study and only hypothesis can be suggested, it is likely that the ageing of the population and also an improved diagnosis of HF during this period, could have an impact on the increased incidence and prevalence of HF.
What could be specific reasons that HF medication is not always prescribed according to guidelines?
There are some reasons why HF medication is not always prescribed according to guidelines. First, underdiagnosis of HF. There is a proportion of patients that are not diagnosed at early stages, and some HF drugs are more difficult to prescribe later.
Barrios V, Escobar C, De La Sierra A, Llisterri JL, González-Segura D. Detection of unrecognized clinical heart failure in elderly hypertensive women attended in primary care setting. Blood Press. 2010;19(5):301-7.
However, therapeutic inertia is the main factor that explains the relative low use of HF medication. Thus, fear of adverse events in frailer patients or advances stages may reduce the use of these drugs or uptitration in some subjects. In addition, some physicians may think that symptoms are more related to medication rather than to HF itself. Low blood pressure or renal insufficiency can be a barrier to the prescription of HF medication use or uptitration. However, a more careful management of these patients, could reduce these potential obstacles. As a result, improving transition of care, a higher use of HF clinics, the development of ambulatory disease management programs, enhancing the role of HF nurses and a higher empowerment of patients and their families could be very helpful to reduce therapeutic inertia, leading to a better prescription of HF drugs.
Girerd N, Von Hunolstein JJ, Pellicori P, Bayés-Genís A, Jaarsma T, Lund LH, Bilbault P, Boivin JM, Chouihed T, Costa J, Eicher JC, Fall E, Kenizou D, Maillier B, Nazeyrollas P, Roul G, Zannad N, Rossignol P, Seronde MF; EF-HF Group. Therapeutic inertia in the pharmacological management of heart failure with reduced ejection fraction. ESC Heart Fail. 2022;9(4):2063-2069.
Are there statistically significant differences between age groups in 2016 and 2019, as might be suspected from the data in the tables?
As 2019 cohort is older than 2016 patients, some comorbidities could be more common in the 2019 cohort. However, the most important fact is that despite the worse clinical profile of 2019 cohort, HF treatment was more optimized.
Has there possibly been a better diagnosis of HF for the Spanish population over the years?
Although there has been an improvement in the diagnosis of HF in Spain in the last years, it remains an important problem in clinical practice.
Anguita M, Bayés-Genís A, Cepeda JM, Cinza S, Cosín J, Crespo M, Egocheaga I, Escobar C, Faraudo M, García-Pinilla JM, Manzano L, Obaya JC, Pascual D, Segovia-Cubero J, Loza E. Expert consensus statement on heart failure with reduced ejection fraction: beyond the guidelines. Rev Esp Cardiol Supl. 2020;20(B):1-46.
Can we tell from the database, how often patients went to the doctor? This could possibly be calculated because of the monthly update. Can anything be said about access to medical care for the Spanish population? Are there differences between urban and rural populations?
This is an important point and in fact, this is the aim of a specific publication. In Spain, the public health system allows free access to the entire population. However, the organization of care for patients with HF may vary between regions (eg availability of HF units, HF nurse, etc.). It is a good point, which requires a specific publication. It has been included as a limitation of the present study.
The sentence in line 146/147 seems oddly worded. Is it really correct like that?
The sentence has been reworded as follows: Quantitative variables were described by measures of central tendency (mean, median) and dispersion (standard deviation, interquartile range).
Overall Escobar and colleagues have presented a manuscript with data of interest for cardiologists, cardiac surgeons, other clinicians, epidemiologists, economists and scientists. It is an excellent work with important findings to understand HF and possible further therapeutic options.
Reviewer 2 Report
Dear Authors,
I would like to thank you for having chance to review the manuscript: " Prevalence, characteristics,management and outcomes of patients with heart failure with preserved, mildly reduced and reduced ejection fraction in Spain". Initially I found the manuscript inquiring but after a while I found some important issues that raise my concern.
The major concern is related to aging population and there is a possibility that increased HF diagnosis is just age-related.
My questions to authors are presented below:
1. study groups were based ICD-10 diagnosis and further on echocardiography results but 8% was not presented with EF, should they be included?
2. the increase in prevalescence of HF in recent times may be related to increased age of the population in Spain. Would it be reasonable to divide the groups HFrEF I HFmEF and HFpEF regarding age in subgroups? The natural history of getting old is related to circulatory disturbances and the trend you present may be related only to this factor.
3. Dear authors, you presented the pharmacotherapy in your groups, but was the therapy changed after initial diagnosis. The scarce informations are so limited that the reader can not draw any further conclusions
4. There is a relatively high percentage of dyslipidemic patients and would you be able to provide initial and obtained results of lipidograms? Was the therapy successfull?
5. The lack of information regarding coronary disease is a significant problem, as well. The information about previous myocardial infarction is provided but more importantly we don't know whether pts was a PCI or CABG history?
6. The pts in HFpEF group are characterized by high prevalence of AF. Was it permanent or paroxysmal? As in the latter, the diagnosis of HF could be given temporary as the new onset of AF occurred and that would change significantly the characteristics of the group
7. The high prevalescence of cancer patients raise another question regarding the etiology of HF. Was it concomitant or related to oncologic therapy (toxic?)
Dear Authors, I hope my small remarks can light a new perspective of the results conclusions if you find them justified.
Kind regards
Author Response
Dear Authors,
I would like to thank you for having chance to review the manuscript: " Prevalence, characteristics,management and outcomes of patients with heart failure with preserved, mildly reduced and reduced ejection fraction in Spain". Initially I found the manuscript inquiring but after a while I found some important issues that raise my concern.
The major concern is related to aging population and there is a possibility that increased HF diagnosis is just age-related.
My questions to authors are presented below:
1. study groups were based ICD-10 diagnosis and further on echocardiography results but 8% was not presented with EF, should they be included?
We respectfully think that this group should be included as in clinical practice it is not uncommon that there are patients with a clinical diagnosis of HF, but without echocardiographic data. It has some interest to ascertain whether the clinical profile and management of these patients differ from the others.
2. the increase in prevalescence of HF in recent times may be related to increased age of the population in Spain. Would it be reasonable to divide the groups HFrEF I HFmEF and HFpEF regarding age in subgroups? The natural history of getting old is related to circulatory disturbances and the trend you present may be related only to this factor.
We think that this is a very interesting point and it is likely that the increased prevalence and incidence of HF may be related with both, the ageing of the population and a better HF diagnosis. However, we consider that this deserves a specific publication. However, we have included a commentary in the text.
3. Dear authors, you presented the pharmacotherapy in your groups, but was the therapy changed after initial diagnosis. The scarce informations are so limited that the reader can not draw any further conclusions
Changes in treatment for HFrEF has been already analyzed and these data have been published in: Sicras-Mainar, A.; Sicras-Navarro, A.; Palacios, B.; Varela, L.; Delgado, J.F. Epidemiology and treatment of heart failure in Spain: the HF-PATHWAYS study. Rev. Esp. Cardiol. 2022, 75, 31-38. This study showed that from the diagnosis (baseline) to 24 months of follow-up, there was discrete treatment optimization. A commentary was included in the manuscript. Further analyses are being conducted in HFpEF recently diagnosed and will be the objective of a separate publication.
4. There is a relatively high percentage of dyslipidemic patients and would you be able to provide initial and obtained results of lipidograms? Was the therapy successfull?
This is also a very interesting point, but we believe this is the objective of a separate research. We think that this study already includes a massive amount of data.
- The lack of information regarding coronary disease is a significant problem, as well. The information about previous myocardial infarction is provided but more importantly we don't know whether pts was a PCI or CABG history?
As we stated in the limitations section, only data that were recorded in the electronic clinical history could be collected, leading to a possible underdiagnosis of some variables. On the other hand, we have information about more variables, but we consider that the information that has been already included in the tables is sufficient to answer the aims of the study. We respectfully think that including more information could be detrimental and distract attention from the focus of the study. - The pts in HFpEF group are characterized by high prevalence of AF. Was it permanent or paroxysmal? As in the latter, the diagnosis of HF could be given temporary as the new onset of AF occurred and that would change significantly the characteristics of the group
As we stated in the limitations section, only data that were recorded in the electronic clinical history could be collected, leading to a possible underdiagnosis of some variables (i.e. tachycardiomyopathy due to AF). - The high prevalence of cancer patients raise another question regarding the etiology of HF. Was it concomitant or related to oncologic therapy (toxic?)
We agree with the reviewer that this is another limitation. We have expanded the limitations section as follows: This was an observational cohort study that used secondary data from electronic health records. Therefore, only data that were recorded in the electronic clinical history could be collected, leading to a possible underdiagnosis of some variables (i.e. tachycardiomyopathy due to AF, cardiotoxicity from cancer treatment, etc.).
Reviewer 3 Report
This article is very important, as it provides information about prevalence, characteristics, management and outcomes of patients with heart failure in Spain using a large database. I have read this article with great pleasure.
However, there are several clarifying questions to the authors.
In the table 1, it is necessary to correct NYHA instead of NHYA (this is probably a typo).
(line 84) The authors pointed out that they excluded patients with chronic kidney disease stage V that required dialysis at any time from the analysis, but in the tables there are data from patients with CKD stage V. That be patients with GFR <15 ml/min/1.73m2? However, they are potential candidates for dialysis in the coming months. The reason for excluding some patients from the study is unclear.
(line 88-90) In my opinion, it is necessary to provide patient data with a description of the symptoms /signs /criteria for the diagnosis of CHF (according to the recommendations 2022 AHA/ACC/HFSA Guideline for the Management of Heart Failure: A Report of the American College of Cardiology/American Heart Association Joint Committee on Clinical Practice Guidelines) and describe in more detail the inclusion of patients in the analysis and distribution into groups. “Although the classic clinical signs and symptoms of HF, together with EF of 41% to 49% or ≥50%, respectively, are necessary for the diagnosis of the HFmrEF and HFpEF, the requirements for additional objective measures of cardiac dysfunction can improve the diagnostic specificity”.
Patients with unspecified ejection fraction and CHF diagnosis raise questions (line 90).
In Tables 1 and 2, an additional division into two subgroups appears among patients with HFpEF (HFpEF (50 to <60) and HFpEF (≥60)), the purpose of this division isn’t explain in the text and informative comments on these data obtained.
In Tables 1 and 2 in the Medications column, it would be very informative to see data on the combination of drugs for the treatment of CHF, since the probability of monotherapy in this cohort is unlikely.
In Tables 1 and 2, “Anemia within 1 year before index date” is recommended to clarify in the explanation to the tables by which criteria it was diagnosed.
(line 136-138), Figure 1 and Figure 2. According to the data presented, there was a decrease in the frequency of CHF in 2019. How could the authors explain this results?
(line208) The Figure 3 presents data “Event Rates in the 2016 HF cohort” . It is probably advisable to provide data for 2019 to understand the dynamics.
The title of the article talks about the outcomes. Are there data on mortality and analysis of patient survival? Since the study contains data from patients who have been under observation since 2013 the authors probably have such information.
Author Response
This article is very important, as it provides information about prevalence, characteristics, management and outcomes of patients with heart failure in Spain using a large database. I have read this article with great pleasure.
Thanks for your comments.
However, there are several clarifying questions to the authors.
In the table 1, it is necessary to correct NYHA instead of NHYA (this is probably a typo).
It has been corrected.
(line 84) The authors pointed out that they excluded patients with chronic kidney disease stage V that required dialysis at any time from the analysis, but in the tables there are data from patients with CKD stage V. That be patients with GFR <15 ml/min/1.73m2? However, they are potential candidates for dialysis in the coming months. The reason for excluding some patients from the study is unclear.
According to the protocol of the study, patients with CKD stage V requiring dialysis any time prior to the index date were excluded from the HF cohort, but not those who developed CKD stage V requiring dialysis during the follow-up. This has been clarified in the text.
(line 88-90) In my opinion, it is necessary to provide patient data with a description of the symptoms /signs /criteria for the diagnosis of CHF (according to the recommendations 2022 AHA/ACC/HFSA Guideline for the Management of Heart Failure: A Report of the American College of Cardiology/American Heart Association Joint Committee on Clinical Practice Guidelines) and describe in more detail the inclusion of patients in the analysis and distribution into groups. “Although the classic clinical signs and symptoms of HF, together with EF of 41% to 49% or ≥50%, respectively, are necessary for the diagnosis of the HFmrEF and HFpEF, the requirements for additional objective measures of cardiac dysfunction can improve the diagnostic specificity”.
Although the idea pointed by the reviewer is of interest, unfortunately this was a database study, using retrospective data, employing inpatient or outpatient HF diagnosis (ICD-10 code) at any time before the index date. This is a limitation of the study that has been included in the manuscript.
Patients with unspecified ejection fraction and CHF diagnosis raise questions (line 90).
Although we could have excluded this group from the analyses, we thought that it would be interesting to include it, as in clinical practice there are patients with a diagnosis of HF, but echocardiographic data are lacking.
In Tables 1 and 2, an additional division into two subgroups appears among patients with HFpEF (HFpEF (50 to <60) and HFpEF (≥60)), the purpose of this division isn’t explain in the text and informative comments on these data obtained.
We performed this analysis, as in the PARAGON-HF trial, the efficacy of sacubitril-valsartan could differ according to LVEF, despite all patients had HFpEF. The same was observed with empagliflozin in EMPEROR-Preserved.
Solomon SD, McMurray JJV, Anand IS, Ge J, Lam CSP, Maggioni AP, Martinez F, Packer M, Pfeffer MA, Pieske B, Redfield MM, Rouleau JL, van Veldhuisen DJ, Zannad F, Zile MR, Desai AS, Claggett B, Jhund PS, Boytsov SA, Comin-Colet J, Cleland J, Düngen HD, Goncalvesova E, Katova T, Kerr Saraiva JF, Lelonek M, Merkely B, Senni M, Shah SJ, Zhou J, Rizkala AR, Gong J, Shi VC, Lefkowitz MP; PARAGON-HF Investigators and Committees. Angiotensin-Neprilysin Inhibition in Heart Failure with Preserved Ejection Fraction. N Engl J Med. 2019;381(17):1609-1620.
Anker, S.D.; Butler, J.; Filippatos, G.; Ferreira, J.P.; Bocchi, E.; Böhm, M.; Brunner-La Rocca, H.P.; Choi, D.J.; Chopra, V.; Chuquiure-Valenzuela, E.; et al. Empagliflozin in Heart Failure with a Preserved Ejection Fraction. N. Engl. J. Med. 2021, 385, 1451-1461.
In Tables 1 and 2 in the Medications column, it would be very informative to see data on the combination of drugs for the treatment of CHF, since the probability of monotherapy in this cohort is unlikely.
Unfortunately, when a patient was taking a fixed combination of 2 or 3 drugs, each drug was considered separately and included in their corresponding category.
In Tables 1 and 2, “Anemia within 1 year before index date” is recommended to clarify in the explanation to the tables by which criteria it was diagnosed.
Regarding the diagnostic codes for anemia, we used the ICD-9 and ICD-10 codes. The supplementary table 1 has been updated. It is important to note that anemia is expected to be underreported as it can be a symptom rather than a diagnosis. This point has been included as a footnote.
(line 136-138), Figure 1 and Figure 2. According to the data presented, there was a decrease in the frequency of CHF in 2019. How could the authors explain this results?
In 2019 the incidence and prevalence of HF decreased, as data were analyzed until September. Although this had been clarified in the text, as follows “Overall, HF event rate was 0.32 per 100 person-years, but increased from 0.27 in 2013 to 0.37 per 100 person-years in 2018 (0.35 per 100 person-years in 2019 -until September-). The overall prevalence of HF was 2.34%, but increased from 2.07% to 2.44%, respectively (2.37% in 2019 -until September-).”, we have now clarified as a footnote in the figures.
(line208) The Figure 3 presents data “Event Rates in the 2016 HF cohort” . It is probably advisable to provide data for 2019 to understand the dynamics.
As data were recorded only until September 2019, we could not provide event rates for this cohort. This is a limitation of the study.
The title of the article talks about the outcomes. Are there data on mortality and analysis of patient survival? Since the study contains data from patients who have been under observation since 2013 the authors probably have such information.
Unfortunately, we have data on mortality only in the incident cohort, but not in the prevalent cohorts. So, this is a limitation of the study.
Reviewer 4 Report
This is a well written study with nice results.The english language level is appropriate. The paper is well structured.
Author Response
Thanks for your comments.
Reviewer 5 Report
This manuscript was designed to estimate the prevalence, incidence, and describe the characteristics and management of patients with heart failure with preserved (HFpEF), mildly reduced (HFmrEF), and reduced ejection fraction (HFrEF) in Spain. The topic is very interesting and the manuscript was well written, so I have just minor comments.
Minor comment
Please provide the meaning of EF in the introduction
Please provide how the sample size was determined in statistical analysis
Author Response
Please provide the meaning of EF in the introduction
The meaning of EF was included in the introduction.
Please provide how the sample size was determined in statistical analysis
No formal sample size calculation was conducted. All patients in the BIG-PAC database meeting the inclusion criteria and with no exclusion criteria were included in the study. This has been added to the statistical analysis section of the manuscript.
Round 2
Reviewer 2 Report
Dear Authors,
I would like to thank you for your corrections of the manuscript but it did not dispel my doubts.
The JCM in a very prestigious journal and the manuscript does not meet the criteria of high scientific value from my humble opinion.
Kind regards
Author Response
Thanks for having reviewed the manuscript, as your comments have been very useful for us to improve the quality of the paper.
Reviewer 3 Report
Thank you for your comments and clarifications to our questions.
Author Response
Thanks